# The Autocrine Role of Placental Extracellular Vesicles from Missed Miscarriage in Causing Senescence: Possible Pathogenesis of Missed Miscarriage

**DOI:** 10.3390/cells11233873

**Published:** 2022-12-01

**Authors:** Yi Zhang, Yunhui Tang, Yang Liu, Jiayi Wang, Ye Shen, Xinyi Sun, Matthew Kang, Min Zhao, Qi Chen

**Affiliations:** 1Department of Obstetrics & Gynaecology, The University of Auckland, Auckland 1141, New Zealand; 2Department of Family Planning, The Hospital of Obstetrics & Gynaecology, Fudan University, Shanghai 200082, China; 3Department of Pathology, Wuxi No. 2 People’s Hospital, Nanjing Medical University, Wuxi 214002, China; 4Department of Family Planning, Wuxi Maternity and Child Health Hospital Affiliated Nanjing Medical University, Wuxi 214002, China; 5Department of Gynaecological Cancer, Wuxi Maternity and Child Health Hospital Affiliated Nanjing Medical University, Wuxi 214002, China

**Keywords:** placenta, placental EVs, miRNAs, senescence, C19MC, missed miscarriage

## Abstract

Placental dysfunction, including senescent changes, is associated with the pathogenesis of missed miscarriage, although the underlying mechanism is unclear. Increasing evidence indicates that placenta-specific miRNAs are packaged in extracellular vesicles (EVs) from placental syncytiotrophoblasts and are released into the maternal circulation. Aberrant cargos including miRNAs in placental EVs have been reported to be associated with the pathogenesis of complicated pregnancies. In this study, we compared the miRNA profiles in EVs derived from missed miscarriage and healthy placentae and investigated possible biological pathways which may be involved in senescence, one cause of missed miscarriage. The total concentration of RNA in placental EVs was not different between the two groups. However, there were 54 and 94 differentially expressed miRNAs in placental large and small EVs from missed miscarriage compared to EVs from healthy controls. The aberrantly expressed miRNAs seen in placental EVs were also observed in missed miscarriage placentae. Gene enrichment analysis showed that some of those differentially expressed miRNAs are involved in cellular senescence, endocytosis, cell cycle and endocrine resistance. Furthermore, transfection of trophoblasts by a single senescence-associated miRNA that was differentially expressed in placental EVs derived from missed miscarriage did not cause trophoblast dysfunction. In contrast, EVs derived from missed miscarriage placenta induced senescent changes in the healthy placenta. Our data suggested that a complex of placental EVs, rather than a few differentially expressed miRNAs in placental EVs derived from missed miscarriage placentae could contribute in an autocrine manner to placental senescence, one of the causes of missed miscarriage.

## 1. Introduction

Missed miscarriage is a subtype of miscarriage that occurs when a fetus is no longer alive or undeveloped but has not been physically miscarried. It affects approximately 1–5% of all pregnancies and 80% of missed miscarriages occur in the first trimester [1]. Chromosomal abnormalities in the fetus are the most common cause of missed miscarriages [2], although other risk factors including advanced maternal age, previous miscarriage, smoking, obesity and diabetes have also been identified [3]. Increasing evidence recently suggested that the dysfunction of placental development including morphological and functional changes in the trophoblasts is also associated with the pathogenesis of miscarriage [4,5,6,7]. Cellular senescence is a state of cell cycle arrest without undergoing cell death, and increased senescence is one of the changes in the placenta that has recently been reported in missed miscarriage [8]. However, the causes of placental senescence are unknown.

It is now well accepted that extracellular vesicles (EVs), lipid-enclosed particles released from all eukaryotic cells studied to date, can facilitate intercellular communication and are classified as large (150 nm to 1 um) and small EVs (10–100 nm) based on their size. During pregnancy, an extraordinary number of large and small EVs are released by the placental syncytiotrophoblast directly into the maternal circulation every day which can be observed as early as the 6th week of gestation [9]. Increasing evidence reports that EVs can be internalized by target cells, including the host cells via multiple mechanisms with the subsequent functional transfer of their cargos, including regulatory RNAs, functional proteins, DNA, and lipids [10]. Abnormal EV production or abnormal packaging of cargos from the placenta, or abnormal uptake by target cells has been suggested to be involved in the pathogenesis of complicated pregnancies such as preeclampsia and gestational diabetes [11,12]. However, the differences of cargos such as miRNAs in placental EVs from missed miscarriage in comparison with healthy pregnancy, and the potential autocrine role of abnormal placental EVs derived from missed miscarriage placenta on placental senescence, has not yet been investigated. It has previously been suggested that cellular senescence can be triggered by physiological stress and molecular changes when a cell is under stress, such as oxidative stress [13].

MicroRNAs (miRNAs) are small single-stranded non-coding RNA molecules that are 22 nucleotides in length. MiRNAs are partially complementary to one or more messenger RNA (mRNA) molecules, and function in RNA silencing and post-transcriptional regulation of gene expression [14]. Compared to other tissues, placental EVs contain high levels of miRNAs that are expressed from the chromosome 19 miRNA cluster (C19MC), which plays a particularly important role in placental development [15,16], and differential expression levels of these miRNAs may contribute to the pathogenesis of complicated pregnancies, including pregnancy loss [17]. Aberrant expression of miRNAs is associated with a number of human diseases by a variety of biological processes (reviewed in [18]).

It is now well-known that EVs contain diverse molecules that are originally derived from the cell of origin [19]. We undertook this study: (1) to investigate the changes in the miRNA profiles in EVs derived from the placentae of missed miscarriage, compared to EVs from healthy placentae; and (2) to investigate whether abnormal placental EVs from missed miscarriage can induce senescent changes in the placenta in an autocrine manner.

## 2. Methods

This study was approved by the Ethics Committee of The Hospital of Obstetrics & Gynaecology of Fudan University, China (reference number 2018/62) and Wuxi Maternity and Child Health Hospital affiliated Nanjing Medical University, Nanjing, China (reference number: 2021/01020204). All placentae were collected with an informed written patient consent form.

### 2.1. Tissue Collection and Preparation

A total of 17 (further details below) placentae from women with missed miscarriage and 17 gestation-matched first trimester placentae from elective surgical termination were collected from the Family Planning Clinic in The Hospital of Obstetrics & Gynaecology of Fudan University and the Department of Family Planning, Wuxi Maternity and Child Health Hospital affiliated Nanjing Medical University, Nanjing, China. The time of uterine curettage in women with missed miscarriage was performed within 7 days, after diagnosis.

### 2.2. Placental EVs Collection and Characterisation

Placental EVs were harvested from placental explant culture as described previously [20]. Briefly, approximately 400 mg of first trimester placental explants (wet weight) were dissected and cultured in Netwell™ culture inserts (400 µm mesh), suspended in 12 well culture plates, at 37 °C in advanced DMEM/F12 containing 2.5% fetal bovine serum (FBS) in an ambient oxygen atmosphere containing 5% carbon dioxide for 24 h. The Netwell™ inserts (containing the explants) were then removed from the culture wells and the conditioned media aspirated for use. Cellular debris was removed from the conditioned media by centrifuging at 2000× *g* for 10 min. The supernatant was then centrifuged at 20,000× *g* for 1 h for placental large EV collection. The supernatant was further centrifuged at 100,000× *g* for 1 h for placental small EV collection (Avanti J30I Ultracentrifuge, JA 30.50 fixed angle rotor, Beckman Coulter, Brea, CA, USA). Morphology of placental EVs was confirmed by electron microscopy and molecular characterisation of CD81 (an EV-enriched marker) and pan-cytokeratin (a marker of placental origin was confirmed by western blot), as previously described [21].

### 2.3. RNA Extraction

The total RNA in the placentae was extracted by adding 1 mL Trizol for 30 min incubation on ice. During the incubation, every 10 min the placental tissues were homogenized using an electric homogenizer. After homogenization, an additional 250 μL of chloroform was added into the solution. The samples were then centrifuged at 12,000× *g* for 15 min at 4 °C after vortexing. After removal of the supernatant, 500 μL of isopropanol was added into the RNA pellet, and then was centrifuged at 12,000× *g* for 10 min at 4 °C. After the centrifuge, the supernatant was discarded, and 75% ethanol (prepared with DEPC water) was added. The RNA solution was then centrifuged at 7500× *g* for 5 min at 4 °C. The RNA pellet was air-dried for 5 to 10 min and dissolved in 10 μL of DEPC water. The concentration of RNA extracted from placentae was measured by Nanodrop at 260 nm (Life Technology, Carlsbad, CA, USA).

### 2.4. RNA Sequencing

RNA in placental EVs or in placentae was extracted by Magzol reagent (Magen, Guangzhou, Guangdong, China) following the manufacturer’s instruction and the RNA concentration was then measured by Qubit (Life Technology, Carlsbad, CA, USA). MiRNA sequencing was conducted by Ribio Co., Ltd. (Shanghai, China). After the extraction of RNA from placental EVs, quality control was conducted by Qubit dsDNA HS Assay Kit (Life Technology, Carlsbad, CA, USA) and RNAScreenTape (Agilent Technology, Santa Clara, CA, USA) using Agilent 2200 TapeStation (Agilent Technology, Santa Clara, CA, USA). Total RNA was fractionated by 15% Tris-borate-EDTA (TBE) polyacrylamide gel (Invitrogen, Shanghai, China), and small RNA (18~30 nt) were gel purified, ligated, reverse transcribed and amplified for cDNA library preparation. Small RNA sequencing was performed by Illumina Hiseq 2500 platform following the Hiseq 2500 User Guide. The RNA sequencing was performed on placental EVs derived from six individual donors (healthy, *n* = 3, and missed miscarriage, *n* = 3).

### 2.5. RNA Sequencing Analysis

The raw data were processed to clean data by the sequencing company, followed by gene mapping and gene annotation via Bowtie and ANNOVAR. The number of miRNAs in EVs reported in this study was defined when the individual miRNA was expressed in at least two individual sequencing measurements from three individual measurements. The definition of expression of miRNAs was the total count value greater than zero. The expression levels of miRNAs reported in this study were calculated as the mean of each Reads Per Million (RPM) value. Differential expression analysis was performed by using Deseq2, with the cut-off value defined as |Log_2_FC| > 1, FDR < 0.05. Gene Ontology (GO) and Kyoto Encyclopaedia of Genes and Genomes (KEGG) enrichment analysis were conducted by using the online miRNAs target gene prediction and enrichment analysis tool miRWalk (http://mirwalk.umm.uni-heidelberg.de, accessed on 1 June 2022).

### 2.6. Measurement of miRNAs in the Placentae by RT-PCR

After extraction of total RNA from the placentae (either missed miscarriage or healthy first trimester placentae), the RNA was reversely transcripted with a stem-loop method using RevertAid^TM^ First Strand cDNA Synthesis Kit (ThermoFisher, Waltham, MA, USA) following the manufacturer’s instruction. The cDNA was then semi-quantified by using the PowerUp SYBR Green MasterMax (ThermoFisher, Waltham, MA, USA) following the manufacturer instruction of the kit on the CFX96 real-time PCR detection system, and the Cq values were calculated and analysed using the relative quantitative method. The expressions of miRNA-125b-1-3p, miRNA512-3p, miRNA-376c-3p, miRNA-376a-3p, miR-377-5p, miR-337-3p and miR-451a that are associated with senescence and highly expressed in EVs were measured. All the primers of these miRNAs were commercially purchased from Ribiobio Co., Ltd. (Shanghai, China).

### 2.7. Transfection of Trophoblast Cells with a Mimic or an Inhibitor of miRNA

Trophoblasts cell line HTR-8/SVneo cells were grown in RPMI 1640 with 10% fetal bovine calf serum (FBS) and 1% streptomycin and penicillin. MiRNA-125b-1-3p mimic and miRNA-512-3p inhibitor were commercially synthesized by Ribobio Co., Ltd. (Shanghai, China), and dissolved in sterile double distilled water to reach the concentration of 50 µM (for miRNA-125b-1-3p mimic) or 100 µM (for miRNA-512-3p inhibitor), following the manufactory’s guidelines. The concentrations of mimic or inhibitor used in this study were more efficient based on the dose response experiment. HTR-8/SVneo cells were seeded into 96 well plates and cultured overnight to reach 50% coverage. After removing the culture medium, the mimic of miRNA-125b-1-3p or inhibitor of miRNA-512-3p or a combination of both were transfected into the cells using the riboEFCT CP transfection kit (RIBOBIO, Shanghai, China) following the manufacturer’s instruction, and cells were further cultured for 24 or 48 h. The cell viability was then measured by Alamar Blue (ThermoFisher, Shanghai, China).

### 2.8. Determine of the Senescent Changes in Placental Explants and Trophoblast Cells

The expression of γH2AX and p21, two markers of senescence, were measured in the healthy first trimester placental explants that had been treated with placental large and small EVs derived from missed miscarriage placentae, by western blotting or immunohistochemistry. In some experiments, the expression of γH2AX and p21 was also measured in HTR-8/SVneo cells that had been transfected with a mimic of miRNA-125b-1-3p or an inhibitor of miRNA-512-3p by western blotting.

### 2.9. Western Blotting

The healthy first trimester placentae were dissected and were cultured in DMEM/F12 medium with 10% FBS and 1% streptomycin and penicillin, in the presence of placental large and small EVs derived from missed miscarriage placentae for 24 h. After protein extraction from placental explants using commercialised RIPA buffer (Applygen, Beijing, China), the concentration of proteins was measured by bicinchoninic acid assay (BCA). The proteins were then denatured by boiling with loading buffer (Cwbio, Beijing, China), followed by electrophoresis on 10% SDS-PAGE gels for 1 h 30 min with the loading of 20 ng samples. Proteins were transferred onto PVDF membranes (0.22 μm) and were blocked with 5% non-fat milk for 1 h at room temperature. After incubation with γH2AX (Abcam, Shanghai, China, 1:1000) or p21 (Abcam, Shanghai, China, 1:1000) primary antibodies at 4 °C overnight, the membranes were then washed with TBS-T and were cultured with secondary antibody (Affinity, Shanghai, China, 1:5000) for 1 h at room temperature. The membranes were treated with ECL (Vazyme, Nanjing, China) and developed on the Chemical Fluorescence luminescence developer.

### 2.10. Immunohistochemistry

The placental explants were, after treatment with placental large and small EVs derived from missed miscarriage placentae, fixed with 4% paraformaldehyde (PFA) for 24 h. The paraffin blocks were made and were then sectioned to 5 µm and the expression of γH2AX and p21 was measured. After being deparaffinized in xylene and rehydrated in graded alcohol, the sections were boiled with citrate buffer (pH = 6.8) for 2 min using a pressure cooker for antigen retrieval. 2.5% horse serum (VectorStain, PK-7800, Singapore, Singapore) was used for blocking and then the sections were incubated with the primary antibody γH2AX (Abcam, Shanghai, China, 1:200) or p21 (Abcam, Shanghai, China, 1:200) at 4 °C overnight. After washing with PBS-T, the sections were incubated with biotinylated pan-specific universal antibody (VectorStain, PK-7800, Singapore, Singapore) for 15 min, and the sections then incubated with streptavidin/peroxidase complex (VectorStain, PK-7800, Singapore, Singapore) for 10 min. DAB kit (Beyotime, Beijing, China) was used for the colour development, and sections were then counterstained for 1 min with haematoxylin and were mounted with mounting medium (Solarbio Life Science, Beijing, China).

### 2.11. Statistical Analysis

The concentration of total RNA was expressed as mean and SD and the levels of miRNAs in placentae were expressed 2^−ΔΔ^Cq. The statistical significantly between the comparisons was assessed using Prison software (version 9.4) with *p* < 0.05.

## 3. Results

### 3.1. The Total Concentration of RNA Associated with Placental Large and Small EVs

The total concentration of RNA associated with placental large and small EVs was not different between EVs collected from healthy first trimester and missed miscarriage placental explant cultures, after adjusting for the weight of placental explants (Figure 1A). The RNA sequencing showed that the proportion of miRNAs in non-coding RNAs in large (Figure 1B, *p* = 0.973) or small EVs (Figure 1C, *p* = 0.388) was not different between the two groups. The proportion of non-coding in total exons in large or small EVs was also not different between the two groups (Data not shown).

### 3.2. Analysis of miRNA Profiles in Placental EVs

Through RNA sequencing, we identified total 1008 miRNAs and 1038 miRNAs associated with placental large EVs derived from healthy and missed miscarriage placentae, respectively. We also identified total 1270 miRNAs and 1250 miRNAs associated with placental small EVs derived from healthy and missed miscarriage placentae, respectively. Although the total number of miRNAs in placental EVs was not different between the two groups, there were 909 miRNAs in large EVs that were commonly expressed in both groups (Figure 2A), while 1120 miRNAs in small EVs were commonly expressed in both groups (Figure 2B). As the expression levels of unique miRNAs in the two groups were lower than 1 in Reads Per Million (RPM), and we focused on comparing the difference in the same miRNAs between the two conditions of pregnancy, we then analyzed the differences in the levels of miRNAs that were commonly expressed in both groups.

### 3.3. Comparison of the Expression Levels of miRNAs between EVs from Healthy and Missed Miscarriage Placentae

We analysed the difference in the expression levels of miRNAs that overlapped between the two groups using Deseq analysis. We found that 54 miRNAs were differentially expressed in the large EVs between the two groups. Of these, 18 miRNAs and 36 miRNAs were upregulated and downregulated, respectively, in large EVs from missed miscarriage placentae, compared to large EVs from healthy placentae (Figure 3A). We also found that there were 94 miRNAs that are differentially expressed in small EVs between the two groups. Of these, 35 miRNAs and 59 miRNAs were upregulated and downregulated, respectively, in small EVs from missed miscarriage placenta (Figure 3B). The differentially expressed miRNAs included in this analysis are listed in Appendix A.

### 3.4. Analysis of the Functions or Signalling Pathways That the Differentially Expressed miRNAs Are Involved in

KEGG analysis showed the top 20 signaling pathways that these differentially expressed miRNAs in large and small EVs are significantly involved in (Appendix A). These signaling pathways include cellular senescence, endocytosis, cell cycle and endocrine resistance. In addition to the top 20 signaling pathways, these differentially expressed miRNAs were also shown to be significantly involved in autophagy, apoptosis and cell adhesion. These three pathways are associated with the pathogenesis of miscarriage [22].

Through GO analysis of biological processes, we showed that some of those differentially expressed miRNAs between the two groups significantly participate in cellular responses to DNA damage and oxidative stress, cellular proliferation and regulation of apoptosis, and response to estrogen production. In addition, those differentially expressed miRNAs are also significantly involved in regulation of extrinsic apoptotic signaling, cellular response to leukemia inhibitory factor and cell growth, which are also associated with the pathogenesis of miscarriage (Appendix A).

### 3.5. Differences in the miRNAs Encoded on Chromosome 19 (C19MC)

The miRNA cluster on chromosome 19 (C19MC) have reported functions on placental development [15,16]. Compared to large EVs derived from the healthy placenta, three miRNAs were downregulated in the C19MC in large EVs derived from missed miscarriage placenta, including miRNA-512-5p, miRNA-519e-5p and miRNA-519b-3p. In addition, compared to small EVs derived from the healthy placenta, there were eight downregulated miRNAs in C19MC in small EVs derived from missed miscarriage placenta, including miRNA-512-5p, miRNA-519e-5p, miRNA-518a-3p, miRNA-520h, miRNA-1323, miRNA-526b-3p, miRNA-515-5p and miRNA-518b.

### 3.6. Changes in the Level of miRNAs That Are Involved in Senescence in Missed Miscarriage Placentae

Our recent study reported that increased senescence was seen in missed miscarriage [8]. It is now well-known that EVs contain diverse molecules that are originally derived from the cell of origin [19]. Thus, we investigated the levels of senescence-associated miRNAs in the placentae of missed miscarriage. The levels of miRNA-125b-1-3p were significantly higher in missed miscarriage placentae than that in age-matched healthy placentae (Figure 4A, *p* = 0.031). In contrast, the levels of miRNA-512-3p, miRNA-376a-3p, miRNA-377-5p, miRNA-337-3p, miRNA-376c-3p and miRNA-451a (Figure 4B–G, *p* < 0.0001) were significantly lower in missed miscarriage placentae than that in age-matched healthy placentae.

### 3.7. Treatment with a Single miRNA Did Not Affect the Growth of Trophoblasts and Did Not Induce the Senescent Changes

To investigate the functions of the aberrant miRNAs on senescence that is associated with the pathogenesis of missed miscarriage [8], HTR-8/SVneo cells were transfected with a mimic of miRNA-125-1-3p, to imitate upregulation, as seen in EVs derived from the placenta of missed miscarriage. HTR-8/SVneo cells were also transfected with an inhibitor of miRNA-512-3p to imitate downregulation, as seen in EVs derived from the placenta of missed miscarriage. The growth of HTR-8/SVneo cells after transfection with a mimic of miRNA-125-1-3p or an inhibitor of miRNA-512-3p or a combination of mimic and inhibitor was not changed after 48 h treatment, compared to untreated cells (Data not shown). In addition, there was no change in the levels of p21 and γH2AX, markers of senescence, in HTR-8/SVneo cells that had been transfected with a mimic of miRNA-125-1-3p, or an inhibitor of miRNA-512-3p, compared to untreated cells (Data not shown).

### 3.8. Placental EVs from Missed Miscarriage Induced Senescent Changes in Normal Placentae

It is well-known that EVs can be re-uptaken by the original cells [23,24]. To understand whether there is an autocrine effect of EVs from the placentae of missed miscarriage on inducing senescence, normal placental explants were treated with EVs derived from the missed miscarriage placentae for 24 h and the senescent changes were measured. The levels of p21 and γH2AX, markers of senescence were significantly increased in healthy placentae that had been treated with large and small EVs derived from missed miscarriage placentae, measured by western blotting (Figure 5) or immunohistochemistry (Figure 6), compared to untreated cells.

## 4. Discussion

In our current study, we observed that there was no difference in the concentration of total RNA and the proportion of miRNAs in non-coding RNA in placental EVs derived from healthy and missed miscarriage placenta. However, there were 54 and 94 differentially expressed miRNAs in placental large and small EVs derived from the placenta of missed miscarriage, compared to EVs derived from healthy placentae, respectively. In addition, there were three and eight miRNAs of the C19MC that were differentially expressed in large and small EVs between the two groups, respectively. A number of senescence-associated miRNAs, including miRNA-125b-1-3p, miRNA-376a-3p, miRNA512-3p and miRNA-199a-3p, were aberrantly expressed in EVs derived from the placentae of missed miscarriage, and in the missed miscarriage placentae. A complex of placental EVs derived from missed miscarriage caused the senescent changes in the placentae.

Placental EVs are lipid bilayer-enclosed packages of cellular contents that are involved in cell-cell communication and signalling, as they carry many functional proteins, regulatory miRNAs, DNA and lipids. The production of EVs is considered to be an evolutionarily conserved process [25,26,27]. A complication of pregnancy may result from abnormal EV production, or abnormal packing of cargos, or abnormal uptake by target cells. In our current study, we found that the total concentration of RNA, and the proportion of miRNAs, and the total number of miRNAs in EVs derived from healthy and missed miscarriage placentae were not different. This could suggest that EVs tend to package any miRNA to fit the total EV volume between the two different conditions of pregnancy. However, by miRNA sequencing analysis, we found that 54 and 94 miRNAs aberrantly expressed in placental EVs from missed miscarriage placentae. Of them, there were 18 upregulated or 36 downregulated miRNAs in placental large EVs, and 35 upregulated or 59 downregulated miRNAs in placental small EVs derived from the placenta of missed miscarriage, compared to EVs derived from healthy placentae. The differences in miRNA expression could be due to the abnormal packaging of cargos in placental EVs by the original cells and consequently contribute to this pathological condition.

Furthermore, using KEGG and GO enrichment analysis, those differentially expressed miRNAs in placental EVs derived from the placenta of missed miscarriage were shown to be involved in pathways of cellular senescence, cell cycle, endocytosis, endocrine resistance and autophagy (Appendix A), and to participate in cellular response to DNA damage and oxidative stress, cell proliferation and regulation of apoptosis, and response to estrogen production (Appendix A). This analysis using KEGG and GO is in agreement with our previous findings that cellular senescence, cellular response to DNA damage and oxidative stress are involved in placental development in missed miscarriage [8].

It is well reported that the contents of cargos in EVs are originally packaged from cells, but this depends on the cell types and conditions of either physiology or pathology [19], and the functions of EVs are dependent on EV cargos [28]. As we previously reported the senescent changes in the placenta of missed miscarriage, we then investigated the levels of senescence-associated miRNAs in the missed miscarriage placentae. From a literature search, we found that miRNA-125b-1-3p, miRNA-512-3p, miRNA-376c-3p, miRNA-376a-3p, miRNA-377-5p, miRNA-337-3p and miRNA-451a are associated with cellular senescence [29,30,31,32,33]. In our current study, we found that the high levels of miRNA-125-1-3b and low levels of miRNA512-5p, miRNA-376a-3p, miR-377-5p, miR-337-3p, miRNA-376c-3p and miR-451 in placental EVs derived from the placenta of missed miscarriage were positively correlated with their respective levels in the missed miscarriage placentae. This confirmed that miRNAs in EVs are originally packaged from the placenta. Although we do not know whether the changes in miRNA profiles analysed in this study are the consequence or the cause of placental senescence, our previous study reported that oxidative stress caused senescence in missed miscarriage [8], and that oxidative stress regulates the expression of miRNAs [34]. This suggests that the aberrant expression of these senescence-associated miRNAs in the placenta could be the consequence of placental senescence in missed miscarriage. Future study is required to investigate whether the changes in miRNA profile are the consequence or the cause of placental senescence.

The MiRNA cluster on chromosome 19 (C19MC), which is highly expressed in placental EVs, has reported functions in placental development [35,36]. These placenta-specific miRNAs are involved in the maternal adaptation during pregnancy [37]. High expression of C19MC miRNAs is associated with the regulation of cell physiology. Abundant levels of these miRNAs have been reported in complicated pregnancies, such as preeclampsia, preterm delivery and fetal growth retardation [15,38,39]. In our current study, we found that three and eight C19MC miRNAs were downregulated in large EVs and small EVs derived from missed miscarriage, respectively. Of them, miRNA-512-5p which is highly expressed in C19MC has been reported to promote cell proliferation and invasion [40], and down-regulated levels of miRNA-512-5p were seen in senescent cells [41]. Taken together, the aberrant levels of C19MC miRNAs reported in our current study may be associated with the dysfunction of placental development seen in missed miscarriage.

It is well-reported that EV cargos can be successfully transferred to the recipient cells leading to functional changes for the recipient cells (review in [42]). To further understand the functions of differentially expressed miRNAs seen in placental EVs derived from the placenta of missed miscarriage, we individually transfected trophoblast cells with a mimic of miRNA-125b-1-3p that was upregulated in placental EVs, or an inhibitor of miRNA-512-3p that was downregulated in placental EVs, or a combination of both. We found that the growth of trophoblast cells was not affected by either miRNA individually or a combination. In addition, individual transfection of miRNA-125b-1-3p mimic or miRNA-512-3p inhibitor with a most efficient dose did not increase the levels of p21 and γH2AX, markers of senescence in trophoblast cells. However, when the healthy first trimester placental explants were treated with placental large and small EVs derived from the placenta of missed miscarriage, the levels of p21 and γH2AX were significantly increased in healthy placentae. Our data suggest that it is unlikely that a single or a few miRNA species in placental EVs contributed to causing senescent changes seen in missed miscarriage placenta. Rather, the manifestation of placental senescence may be produced through a complex of abnormal EVs. Although studies have suggested that a single miRNA, such as miRNA-517a in placental EVs, could regulate the maternal immune system [43], other studies have reported that endoplasmic reticulum stress stimulated the production of trophoblastic EVs that are associated with systemic maternal inflammatory response [44]. Our data also suggests that there is an autocrine role of placental EVs from missed miscarriage causing senescence, as we found that a number of senescence-associated miRNAs were highly expressed in EVs, and EVs can be re-uptaken by the original cells [23,24].

There are some limitations in our current study. We did not investigate the potential underlying mechanism of increased senescence in the healthy placentae induced by EVs derived from missed miscarriage. However, including miRNA sequencing analysis presented in this study, our proteome of placental EVs also shows that there are some highly expressed proteins in placental EVs that are associated with inducing senescence (http://www.iprox.cn (accessed on 25 May 2021). Accession number: IPX0002999003). This suggests that a complex of placental EVs could impact the functions of placental cells in an autocrine manner.

## 5. Conclusions

Our current study demonstrates that there was a number of differentially expressed miRNAs in placental EVs derived from missed miscarriage placentae, including C19MC miRNAs and senescence-associated miRNAs. In addition, the abnormal placental EVs produced from the placenta of missed miscarriage subsequently induced the senescent changes in the placenta, suggesting an autocrine role of abnormal placental EVs in the pathogenesis of missed miscarriage.

## Figures and Tables

**Figure 1 cells-11-03873-f001:**
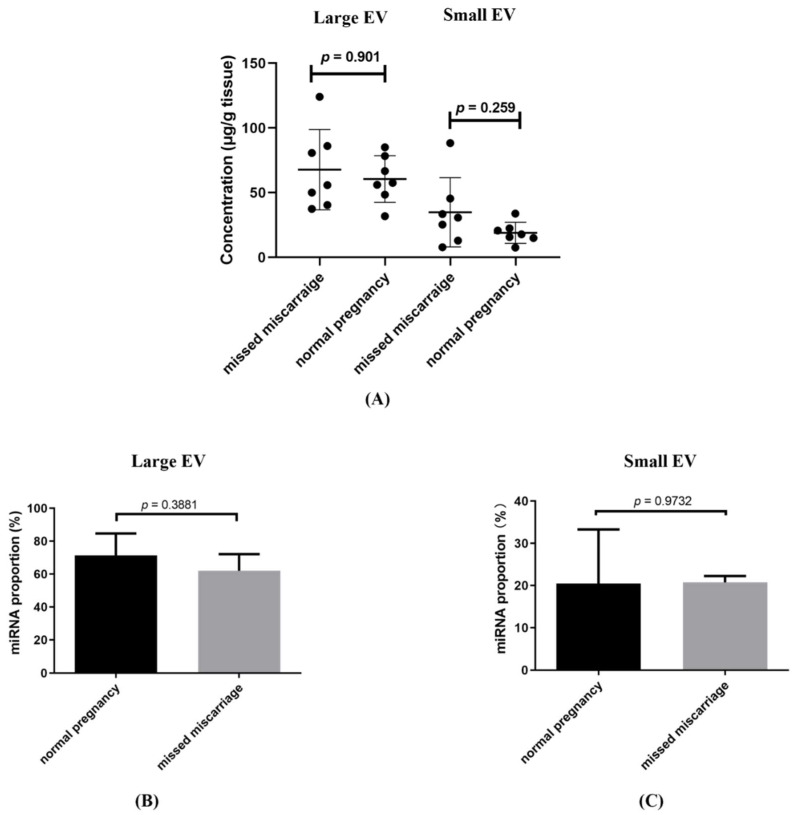
The concentration of total RNA associated with placental large and small EVs derived from healthy or missed miscarriage placentae (**A**), the proportion of miRNAs in non-coding RNAs in large EV (**B**) and small EVs (**C**) (data represents mean ± SD with seven repeats).

**Figure 2 cells-11-03873-f002:**
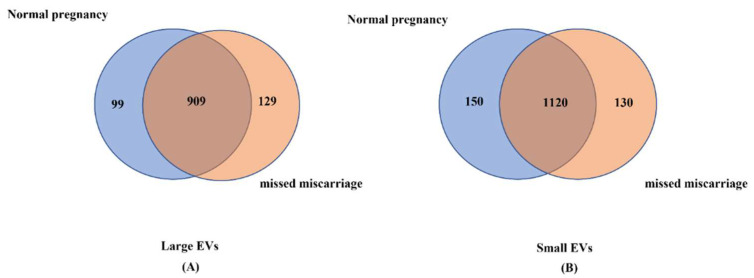
The total number of miRNAs in placental large EVs (**A**) and small EVs (**B**) between EVs derived from healthy or missed miscarriage placentae.

**Figure 3 cells-11-03873-f003:**
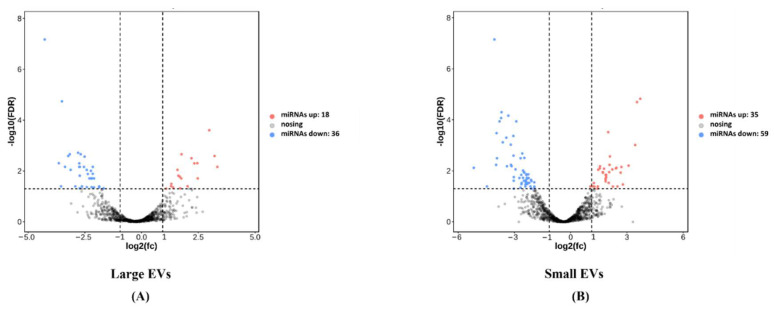
The volcano maps show the numbers of the abundantly expressed miRNAs in placental large EVs (**A**) and small EVs (**B**) derived from missed miscarriage placentae.

**Figure 4 cells-11-03873-f004:**
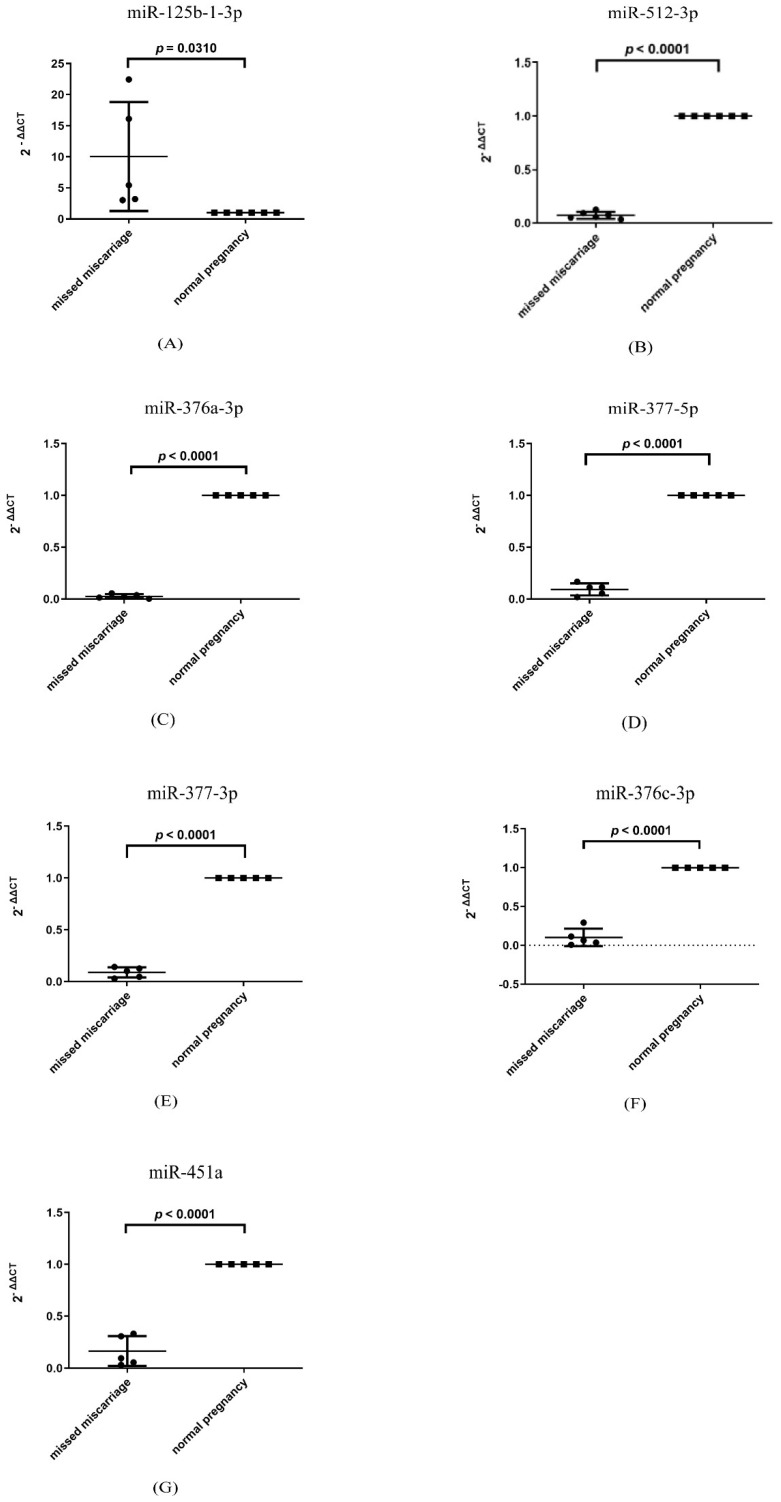
(**A**–**G**): The levels of senescence-associated miRNAs in missed miscarriage placentae (data represents mean ± SD with five repeats).

**Figure 5 cells-11-03873-f005:**
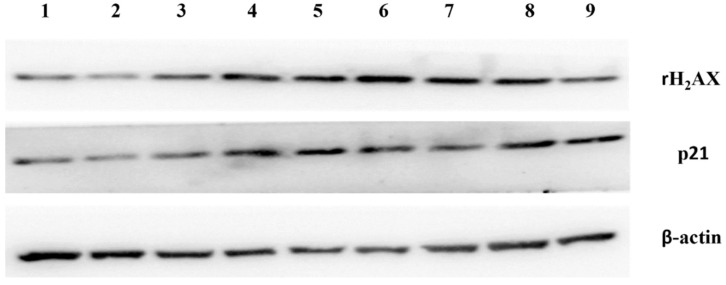
Western blotting showing increased levels of γH2AX and p21 in healthy first trimester placental explants that had been treated with large and small EVs derived from missed miscarriage placentae (1–3: controls; 4–6: large EVs treated; 7–9: small EVs treated).

**Figure 6 cells-11-03873-f006:**
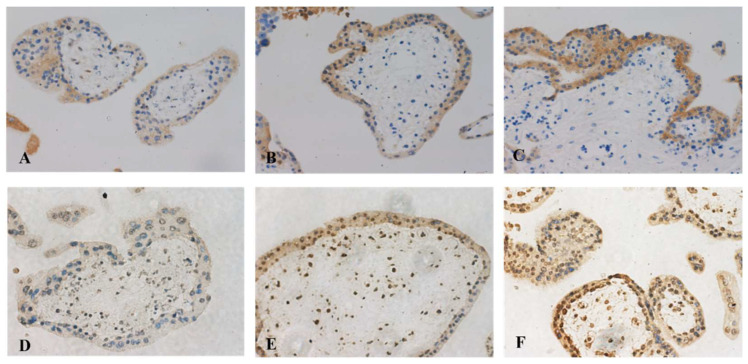
Representative images of immunohistochemistry showing increased levels of γH2AX (**A**–**C**) and p21 (**D**–**F**) in healthy placental explants that had been treated with large EVs (**B**,**E**) and small EVs (**C**,**F**) derived from missed miscarriage placentae, compared to untreated cells (**A**,**D**). (magnification: 400×).

## Data Availability

The datasets used and/or analysed during the current study available from the corresponding author on reasonable request. The raw miRNA sequencing data included in this study has been uploaded on the European Nucleotide Archive (http://www.ebi.ac.uk/ena/data/view/PRJEB44935 (accessed on 12 July 2022), accession number: PRJEB44935).

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
