# Peer review of "The Autocrine Role of Placental Extracellular Vesicles from Missed Miscarriage in Causing Senescence: Possible Pathogenesis of Missed Miscarriage"

_cells, 2022, doi:10.3390/cells11233873_

Round 1

Reviewer 1 Report

The research entitled “The autocrine role of placental extracellular vesicles from missed miscarriage in causing senescence: possible pathogenesis of missed miscarriage” authored by Zhang et al., is very interesting work. Authors investigated the potential roles of extracellular vesicles and its association with missed miscarriage. This type of research might provide us more details about the unexplored roles of extracellular vesicles in different normal and pathogenesis status. I recommend for publication after minor revision, I attached my comments with pdf in the original paper.

Figure 4, should be separated into subfigure,4a, 4b, …etc. (I think its better to make the same figure 6).

Author Response

The research entitled “The autocrine role of placental extracellular vesicles from missed miscarriage in causing senescence: possible pathogenesis of missed miscarriage” authored by Zhang et al., is very interesting work. Authors investigated the potential roles of extracellular vesicles and its association with missed miscarriage. This type of research might provide us more details about the unexplored roles of extracellular vesicles in different normal and pathogenesis status. I recommend for publication after minor revision, I attached my comments with pdf in the original paper.

Answer: Many thanks for your positive comments.  We have made the changes in the main text as suggested.

Figure 4, should be separated into subfigure,4a, 4b, …etc. (I think its better to make the same figure 6).

Answer: we have now made the changes as suggested.

Reviewer 2 Report

This article is interesting in that it evaluated the pathogenesis of early miscarriage with a focus on cellular senescence.

The authors performed comparative validations between miscarriage samples and healthy control samples, but this study design inevitably cannot avoid the consequences of the outcome of cell death due to miscarriage. It is likely that the signal (e.g., miRNA expressions) reflected the outcome rather than the cause of the miscarriage. In the miscarriage samples, the time between miscarriage and D&E was also unknown. I think this is a weak explanation for the mechanism of miscarriage.

And, the authors also analyzed miRNAs commonly expressed in miscarriage and health control samples, but could miRNAs that were not overlapped also play an important role? The reason for excluding non-overlapped miRNAs is not clear.

It would also be better to indicate whether the collected Large EV and Small EV specifically expressed each marker.

The letters D, E, and F in Figure 6 are not listed.

November 24: I confirmed the revised manuscript, although once I rejected it. I understand the reply to my comments. Final recommendation: accept.

Author Response

This article is interesting in that it evaluated the pathogenesis of early miscarriage with a focus on cellular senescence.

Answer: Many thanks for your positive comments. 

The authors performed comparative validations between miscarriage samples and healthy control samples, but this study design inevitably cannot avoid the consequences of the outcome of cell death due to miscarriage. It is likely that the signal (e.g., miRNA expressions) reflected the outcome rather than the cause of the miscarriage.

Answer: We do agree that the changes in miRNA profiles analysed in this study could be the consequence or the cause of placental dysfunction, such as senescent changes. We have previously reported that senescent changes in placenta induced by oxidative stress is one of the causes of miscarriage [5]. It is well-known that oxidative stress regulates the expression of miRNAs. In the present study, we compared the miRNA profiles between the two conditions of pregnancy. In addition to the senescent changes in placenta, importantly, in the present study we found a complex of placental EVs derived from missed miscarriage placentae could contribute in an autocrine manner to placental senescence, suggesting a negative feedback in causing placental dysfunction. Therefore, our present study suggests another potential mechanism for the pathogenesis of missed miscarriage. We have now modified our sentences to make this more clearly (page 16-17). However, future study is required to investigate whether the changes in miRNA profile is the consequence or the cause of placental senescence.

In the miscarriage samples, the time between miscarriage and D&E was also unknown. I think this is a weak explanation for the mechanism of miscarriage.

Answer: The time of uterine curettage was performed within 7 days, after diagnosis. We have now added these information in the methods section.

And, the authors also analyzed miRNAs commonly expressed in miscarriage and health control samples, but could miRNAs that were not overlapped also play an important role? The reason for excluding non-overlapped miRNAs is not clear.

Answer: As the expression levels of unique miRNAs in the two groups were lower than 1 in Reads Per Million (RPM) and we focused on comparing the difference in the same miRNAs between the two conditions of pregnancy, we then analysed the differences in the levels of miRNAs that were commonly expressed in both groups. We have now modified our sentences (page 11).

It would also be better to indicate whether the collected Large EV and Small EV specifically expressed each marker.

Answer: to date there is no specific marker for large and small EVs. All the markers are expressed in both large and small EVs.

The letters D, E, and F in Figure 6 are not listed.

Answer: the missing letter was due to error when the word file was converted to the PDF file. We have now uploaded the word file for the figures.

Reviewer 3 Report

​I reviewed the manuscript and I think it is acceptable for publishing.

Reviewer 4 Report

The study by Yi Zhang et al., has the merit to broaden our knowledge of possible mechanisms leading to missed miscarriages, by taking into considerations the possible role of miRNAs packaged into EVs. Although a clear underpinning of the relationship between placental senescence and differentially expressed miRNAs has not been cleared, the study is still worth being published and exposed to other specialists in the field.